

# Seasonality and alternative floral resources affect reproductive success of the alfalfa leafcutting bee, *Megachile rotundata*

Casey M. Delphia[1,2], Laura A. Burkle[3], Joshua M. Botti-Anderson[2,4] and Kevin M. O'Neill[2]

[1] Montana Entomology Collection, Montana State University, Bozeman, Montana, United States
[2] Department of Land Resources and Environmental Sciences, Montana State University, Bozeman, Montana, United States
[3] Ecology Department, Montana State University, Bozeman, Montana, United States
[4] Fenner School of Environment and Society, Australian National University, Acton, Australian Capital Territory, Australia

## ABSTRACT

**Background:** Managed populations of the alfalfa leafcutting bee (ALCB), *Megachile rotundata* (F.), are often not sustainable. In addition to numerous mortality factors that contribute to this, the dense bee populations used to maximize alfalfa pollination quickly deplete floral resources available to bees later in the summer. Providing alternative floral resources as alfalfa declines may help to improve ALCB reproduction.

**Methods:** We examined the relationship between floral resource availability and ALCB reproduction and offspring condition *via* (1) a field study using alfalfa plots with and without late-blooming wildflower strips to supply food beyond alfalfa bloom, and (2) a field-cage study in which we provided bees with alfalfa, wildflowers, or both as food resources.

**Results:** In the field study, bee cell production closely followed alfalfa floral density with an initial peak followed by large declines prior to wildflower bloom. Few bees visited wildflower strips, whose presence or absence was not associated with any measure of bee reproduction. However, we found that female offspring from cells provisioned earlier in the season, when alfalfa predominated as a source of provisions, eclosed with greater body sizes and proportion body lipids relative to total body mass. For bees restricted to cages, the proportion of offspring that survived to adults was highest on pure alfalfa diets. Adding wildflowers to cages with alfalfa did not affect adult offspring production or female offspring body size and lipid content. Furthermore, although similar numbers of adults were produced on wildflowers alone as with alfalfa alone, females eclosed with smaller body sizes and lower proportion body lipids on wildflowers despite the higher protein content we estimated for wildflower pollen. We found no evidence that adding the late-season wildflower species that we chose to plant enhanced ALCB offspring numbers. Our results highlight the importance of considering multiple measures of reproductive success, including offspring body size and lipid stores, when designing and evaluating floral resource management strategies for agroecosystems.

Corresponding author
Casey M. Delphia,
casey.delphia@montana.edu

# INTRODUCTION

The alfalfa leafcutting bee (ALCB), *Megachile rotundata* (F.) (Hymenoptera: Megachilidae), is one of the most economically important managed bees in North America, particularly for its use as the major pollinator of seed alfalfa (*Medicago sativa* L.) (*Pitts-Singer & Cane, 2011*). ALCB is a solitary bee whose females construct linear series of brood cells within tunnels, including artificial versions provided for management. Each cell is lined by the female with cut leaf pieces and provisioned with pollen and nectar for a single larva. Alfalfa seed farmers rear adult bees for release each season from brood cells produced the previous year. Bee population losses during growing seasons and overwintering result from multiple causes often related to management practices, including diseases like chalkbrood, parasitoids and predators, dispersal, and thermal stresses (*James & Pitts-Singer, 2013*; *Donahoo et al., 2021*). In the U.S., fewer than 50% of bees released for pollination are replaced during reproduction, requiring growers to import costly bees from Canada in some years (*Pitts-Singer, 2008*; *Pitts-Singer & Cane, 2011*). Thus, improved management strategies are needed that promote the long-term health and sustainability of this pollination system.

Most research aimed at increasing the sustainability of ALCB populations has focused on the management of natural enemies (*e.g.*, *Whitfield & Richards, 1985*), mitigation of disease (*e.g.*, *James, 2005*), and optimizing rearing strategies (*e.g.*, *Richards, Whitfield & Schaalje, 1987*; *Pitts-Singer & James, 2009*; *Yocum et al., 2010*; *O'Neill et al., 2011*). Another important consideration for sustaining bee populations is that bee reproductive success also depends on the quality and quantity of available floral resources (*Roulston & Cane, 2000*). Typical ALCB management in the U.S. involves releasing high densities of bees to increase alfalfa seed set, however, this practice quickly depletes floral resources (*Strickler & Freitas, 1999*; *Pitts-Singer, 2008*, *2013*), limiting late-season brood production. Thus, management decisions can result in tradeoffs between seed yield and bee reproduction (*Strickler, 1996*).

One potential strategy to support pollinators in agroecosystems is to plant alternative, non-crop flowering plants to enhance and/or extend floral availability beyond the short-lived bloom of monoculture crops (*e.g.*, *Blaauw & Isaacs, 2014*; *Williams et al., 2015*; *Burkle, Delphia & O'Neill, 2020*; *Klatt, Nilsson & Smith, 2020*; *Graham et al., 2020*). In alfalfa, as floral resources decline in late summer, female bees may be forced to fly farther afield. This decreases their foraging efficiency and increases their own energy needs, resulting in production of a higher ratio of sons to the daughters (*Peterson & Roitberg, 2006a*) that are the next year's primary pollinators. Additionally, when faced with lower alfalfa floral resource levels, they produce fewer and smaller female offspring (*Peterson & Roitberg, 2006b*), reducing not only their own reproductive success but potentially that of the following generation because smaller females produce smaller eggs (*O'Neill, Delphia & O'Neill, 2014*). Adding wildflower strips that provide an alternative source of pollen and

nectar might help alleviate this problem, provided those species (1) flower after peak alfalfa bloom so as not to impact alfalfa pollination *via* competition for pollinators (*Lander et al., 2011*; *Nicholson et al., 2019*) and (2) do not decrease alfalfa seed purity levels should plants later spread into alfalfa fields.

Wildflower plantings have the potential to provide food resources to ALCBs because females are known to forage on a wide variety of plant families (*Small et al., 1997*). A study in Montana found that non-alfalfa pollen comprised over 40% of the pollen grains carried by females from mid-August to early September (*O'Neill & O'Neill, 2011*), during a time when fewer alfalfa flowers are present. These alternative pollens came from diverse weed species from five plant families (Asteraceae, Brassicaceae, Chenopodaceae, Fabaceae, and Scrophulariaceae) that grew within or adjacent to alfalfa fields; yellow sweetclover (*Melilotus officinalis* (L.) Lam.) and a mustard species (Brassicaceae) were the two most abundant pollens collected. However, the potential value of these species as food resources for bees in alfalfa seed fields conflicts with weed management practices. Therefore, the introduction of late-blooming, non-weed flowering plants seems promising.

Using supplemental floral resources to extend the seasonal availability of pollen and nectar in alfalfa seed fields could lead to greater numbers of offspring, and/or larger and healthier offspring (*Torchio, 1985*; *Minckley et al., 1994*; *Strickler & Freitas, 1999*), provided that the alternative pollens are of sufficient nutritional value (*Vaudo, Dyer & Leonard, 2024*). In solitary bees, body size is influenced by the amount of pollen and nectar they receive as larvae from their mothers (*Klostermeyer, Mech & Rasmussen, 1973*; *Roulston & Cane, 2000*). However, adult female condition has been shown to decrease later in the summer (*O'Neill, Delphia & Pitts-Singer, 2015*) and can also influence offspring body size and sex ratios due to reduced provisioning efficiency (*Sugiura & Maeta, 1989*; *Seidelmann, 2006*). Previous studies indicate that larger ALCB females start the nesting season with proportionally greater lipid stores, a likely indicator of offspring condition (*O'Neill, Delphia & O'Neill, 2014*). These lipid stores decline by about a third within 1 week after adult females begin nesting activity in fields, likely because they are used to produce eggs, which also decline in size as the nesting season progresses (*O'Neill, Delphia & Pitts-Singer, 2015*). These and other age-related declines in the condition of adult bees (*e.g.*, *O'Neill, Delphia & Pitts-Singer, 2015*) could negate the potential benefits of additional food resources if females are unable to capitalize on their availability. Alternatively, providing food resources late in the summer could support newly emerging 2nd generation bees, which are common in seed production fields in many locations in the western U.S. (*Johansen & Eves, 1973*; *Pitts-Singer & Cane, 2011*). Whether late-season supplemental floral resources (and the quality of those resources) can enhance bee reproduction or offspring condition (*e.g.*, body size or lipid stores) was the hypothesis that we aimed to test.

To explore the potential benefits of using wildflower strips for increasing ALCB reproductive success, we established alfalfa field plots with and without adjacent flower strips on a research farm in southcentral Montana, USA. Our objectives were to evaluate, during two field seasons, the effects of adding floral resources that bloom after peak alfalfa bloom on (1) ALCB foraging behavior, (2) ALCB reproductive success (including offspring production, survival, and condition), and (3) alfalfa seed yields. We hypothesized that

addition of wildflower strips that flower after peak alfalfa bloom would lead to increases in ALCB offspring production without negatively impacting alfalfa seed yields. Our experimental design also allowed us to examine seasonal changes in offspring condition over the course of the nesting season. Due to unavoidable issues with flowering phenology and drought that limited our ability to fully address our initial objectives, we report results from one summer of the field study, along with a field-cage study that aimed to overcome some of the limitations of the open-field design. In the field-cage study, nesting bees were provided either alfalfa, wildflowers, or a combination of the two. We modified and extended our objectives to evaluate the effects of alternative floral resources on (1) ALCB reproductive success and (2) ALCB pollen provisioning, as well as (3) the effect of provision quality on offspring condition.

## MATERIALS AND METHODS

*Study site.* This research was conducted in southwest Montana at Lutz Farm (45.8132°N, 111.0550°W), an Agricultural Experiment Station operated by Montana State University (MSU) north of Bozeman, MT. The farm is 650 acres (ca. 0.80 × 2.83 km) and broken up into smaller fields of varying sizes planted primarily with wheat, barley, and other small grains. Because the surrounding crops were primarily wind-pollinated, it limited available floral resources for bees to those we provided in our experimental design.

**Field study:** Do late-season supplemental wildflower resources enhance ALCB reproduction and offspring condition?

*Alfalfa plots.* In early May 2016 we established two 0.05-ha (23 m × 23 m) alfalfa plots (variety 'Cooper', Seed Source, Inc., Toston, MT) at each of six sites (distributed in a 2 × 3 grid) at Lutz farm. Sites were at least 400 m apart to reduce the likelihood that bees would move between them (*Tepedino, 1983*; *Gathmann & Tscharntke, 2002*). To hasten blooming for sampling in year 1 (2016), we transplanted alfalfa plugs grown in the greenhouse in late winter into one randomly selected plot at each site. We hand-seeded the second plot for sampling in year 2 (2017). All alfalfa (plugs and seeds) were planted on 0.91 m centers to ensure plants received adequate ground moisture as there was no irrigation (*Mueller, 2008*).

In year 1, alfalfa bloom was delayed by 3 weeks, likely due to transplanting, and overlapped entirely with wildflower strip bloom, which was not our intended experimental design; data from year 1 are not reported in this study. In year 2, alfalfa bloomed earlier than in year 1, but then declined quickly when the site experienced the second driest August on record. That year the wildflower strips bloomed after peak alfalfa bloom as we planned, but most bee activity had ceased once bloom in wildflower strips began to peak, affecting what relevant data we could collect. Despite these difficulties, we obtained useful information on how offspring fitness was correlated with dates of brood cell provisioning during year 2.

*Wildflower strips.* We selected nine wildflower species for the flower strips: common marigold (*Calendula officinalis* L.), deerhorn clarkia (*Clarkia pulchella* Pursh), Rocky Mountain beeplant (*Cleome serrulata* Pursh), plains coreopsis (*Coreopsis tinctoria* Nutt.), garden cosmos (*Cosmos bipinnatus* Cav.), common sunflower (*Helianthus annuus* L.),

showy goldeneye (*Heliomeris multiflora* Nutt.), desertbells (*Phacelia campanularia* A. Gray), and lacy phacelia (*Phacelia tanacetifolia* Benth.). The species list includes primarily native annuals, one native perennial that blooms from seed in its first year, and two non-native annuals. We chose these plant species because: (1) annuals establish quickly and bloom in the same year they are seeded, allowing more control over the timing of flowering by altering seeding dates; (2) native plants are a good ecological fit with the natural landscape and could also provide added value by supporting native bee populations; (3) a diversity of flowering plants with different flower colors and morphologies provides varied pollen and nectar sources, creating a mixed diet for bees; and (4) these species are reasonably easy to grow, have commercially available and inexpensive seed, and can grow in full sun with reasonable drought tolerance.

To help guide our plant selection, we also took into consideration (1) published records of *M. rotundata* visitation to plants (*Stubbs, Drummond & Osgood, 1994*; *Jensen, O'Neill & Lavin, 2003*; *O'Neill et al., 2004*; *Cane, 2008*; *Teper, 2008*; *O'Neill & O'Neill, 2011*); (2) species used in restoration seed mixes for rehabilitation of western rangelands (*Herron et al., 2013*); (3) species being used in insectary plant mixtures (*Grasswitz, 2013*); (4) species being tested for creating "bee pastures" for increasing bee populations (*Cane, 2010*); and (5) species on which bees will readily forage but that *do not* produce seeds that are similar in size and shape to alfalfa seeds which would not be suitable for growing near alfalfa fields because escaped plants could become weedy in alfalfa seed fields and/or decrease alfalfa seed yield purity, like yellow sweet clover, *M. officinalis*.

Half of the six sites received the flower strip treatment, alternating based on site location so that adjacent sites received different treatments. In mid-June of 2017, we seeded wildflower strips (4.5 × 25 m) alongside the alfalfa plots. Each wildflower strip contained three, 0.61 × 4.5 m-replicates of each of the nine species for a total of 27 subplots per strip with a 0.30-m walking path between subplots.

***Bee nest shelters.*** We constructed six wood nesting shelters (designed after *O'Neill, 2004*; Fig. S1) and placed one shelter alongside the edge of each alfalfa plot. Wood-laminate nesting blocks were placed in each shelter for bees to nest (for more details see Fig. S1). Shelters were oriented so that (1) the opening of the shelter faced the alfalfa plot and (2) the flower strips were perpendicular and equidistant to the shelters (12 m from one end of the flower strip) in all plots. This meant that the cardinal direction each shelter opening faced varied based on the layout of the field at each site such that shelters faced either north, northeast, northwest, or east (field layout was determined by the available space at the research farm to accommodate our plots). Shelter orientation was examined as a potential covariate for analyses.

***Alfalfa leafcutting bee management.*** In early February, we purchased alfalfa leafcutting bee cells (*i.e.*, brood cells) from a local alfalfa seed grower (Seed Source, Inc., Toston, MT, USA) containing overwintering ALCB prepupae and placed them into cold storage (6 °C) until needed. Three weeks before alfalfa was expected to be 25–50% full bloom, we removed the bee cells from cold storage and divided them by weight into six trays. We kept them at room temperature for 24 h to acclimate before placing them in a growth chamber in early June at 28 °C to initiate their development into adults for field release. On 3 July

2017, when adult females began to emerge, we moved them to the field shelters in the alfalfa plots. We followed standard recommendations for the quantities of bees released for alfalfa seed pollination in the U.S. (*i.e.*, 40,000–60,000 bees/acre; *Pitts-Singer & Bosch, 2010*); we set out ca. 6,000 bee cells per plot. Prior to releasing bees, we reared a subset of ca. 550 bee cells which resulted in adult emergence success of 53%, with 43% females for an estimated 1,367 adult females released per plot.

*Alfalfa and wildflower floral density.* We measured alfalfa floral density for each plot each week after bee release at 0730–1100 h. We divided each plot into thirds, and along one linear transect within each third (each chosen using a random number generator), we estimated floral density within a circular subplot (491 cm$^2$) at three points chosen using a random number generator for both the distance into the row and the position within each row (left or right side of row). Following methods similar to *Pitts-Singer (2013)*, we counted (1) the total number of racemes within the subplot and (2) open flowers classified as tripped (pollinated) and untripped (still available as a food resources) on three racemes that were closest to three evenly spaced markings on the hoop. The "tripped" category included flowers that were "newly tripped" (tripped that day) and "old tripped" (likely tripped the previous day or 2 days). We calculated the average number of open flowers per subplot as: number of open flowers counted per raceme × number of racemes counted in the subplot. We then used this information to estimate floral density in the entire plot. In general, alfalfa flowers were evenly distributed throughout the plots due to the way in which we spaced the plants at the start of the study. We used these same calculations for tripped flowers. For our purposes we added the open and tripped (newly tripped and old tripped) flowers together for floral density estimates because this represented the food resources available to bees within that week. Alfalfa floral density was used to understand the timing of bloom relative to wildflower strips and examined as a potential covariate for analyses.

We also measured floral density of each species within each wildflower strip weekly from the onset of bloom to understand the density and timing of wildflower bloom relative to alfalfa bloom in each plot. Each of the 27 wildflower strip subplots was divided in thirds or in half, depending on the floral density, and all open flowers were counted then multiplied by 2 or 3, depending on how the subplot was divided, to estimate total flowers for each subplot. One species, *C. serrulata*, did not germinate, and another, *C. pulchella*, produced very few blooms.

*Flower visitation and pollen load composition.* To examine flower visitation, we conducted weekly, timed observations of ALCB foraging in wildflower strips and surrounding weedy vegetation. All observations were conducted on calm, sunny days during 1130–1600 h. For wildflower strips, we conducted 3-min timed observations at each subplot (9 min total per plant species per plot) and recorded the abundance and sex of ALCBs visiting flowers. For surrounding vegetation, we conducted 3-min timed observations on each plant species within a 50-m radius of the alfalfa plot and recorded the abundance and sex of ALCBs visiting the flowers. The surrounding vegetation included 10 species total from the genera *Liatris*, *Tanacetum*, *Solidago*, *Helianthus*, *Berteroa*, *Lupinus*, *Clematis*, *Achillea*, *Heterotheca*, and *Chamaenerion*. All species were distributed in small

patches that ranged from an individual plant to a handful of plants in areas no larger than ca. 2 m$^2$. For all observations, ALCBs were visually noted and recorded. Because bees were not individually marked, we could not avoid recording the same individual, so it is possible that the same individual could have been recorded more than once. But the duration of the observations was short, and we never saw more than three bees per observation period.

To examine bee pollen load composition, at each plot each week, we captured ca. 10 female bees returning to their nests with full pollen loads, placed them in vials, chilled them on ice, and then removed the pollen loads using dental toothpicks (The Doctor's BrushPicks®). Following the methods of *O'Neill et al. (2004)*, pollen from bees was stained using Safranin O solution (Home Science Tools, Billings, MT, USA) and fixed on microscope slides for examination using light microscopy. We identified the pollen collected from bees by comparing them to our pollen reference collection comprised of samples of pollen collected from plant species that bloomed that season; all pollen identifications were conducted by the same individual using the reference collection we created consisting of 24 co-flowering species from five plant families (Asteraceae, Boraginaceae, Fabaceae, Onagraceae, and Ranunculaceae). Pollen grains from taxa not represented by our reference collection were classified as morphospecies and reference photos were taken as they were encountered to serve as a digital reference collection. We calculated the percentage of pollen grains of each plant species in each pollen load based on at least 100 grains per slide; percentages were not corrected for size of the pollen grains. Because of concerns with the identification of *Phacelia* pollen from the flower strips (see Results), we provide descriptive data on percentages of alfalfa *vs*. non-alfalfa pollens in pollen loads only; no statistical analyses were performed.

***Offspring production and survival.*** To determine approximate timing of when nests were occupied and cells were provisioned with pollen, we conducted weekly observations at all plots and marked the end cap of newly completed nests in each shelter with a different color of paint pen for each week.

In late September, we removed all nesting blocks from shelters. For each plot, we quantified the total numbers of completed nests per plot in each week based on their color paint markings. We then carefully separated individual bee cells from all completed nests and (1) weighed, as a group for each week, all the cells for each plot and (2) determined the average weight of 100 cells for each plot each week. We used these measures to estimate the total number of bee cells produced for each plot each week. Bee cells were then placed into cold storage (6 °C) for overwintering until they were removed the following spring to initiate their development into adults for offspring survival and fitness measures.

In spring 2018, we removed from cold storage a subset of bee cells produced during the previous field season and reared offspring to adults in a growth chamber at 28 °C. We reared bees in individual gelatin capsules (#000) with a hole punched in each end for air exchange; for each plot we chose 80 to 250 cells, depending on availability, provisioned during each of 8 weeks. Each day we checked for bee emergences and immediately freeze-killed emerged adults for examining offspring condition. After bee emergences ceased, we counted the total number of emerged adults and calculated emergence success of offspring (measured as the proportion of adults emerging from all cells containing

offspring). We dissected all cells from which no adults emerged to determine whether the cells contained offspring and, if so, the stage reached by the offspring prior to death (*e.g.*, egg, larva, prepupae, pupa, or unemerged adult). For each plot in each week, we calculated the proportion of offspring that died at each stage, hereafter offspring mortality.

***Offspring condition.*** To evaluate offspring quality, we measured body lipid content and body size of adult female offspring that emerged following methods previously used for this species (*O'Neill et al., 2011*; *O'Neill, Delphia & O'Neill, 2014*; *O'Neill, Delphia & Pitts-Singer, 2015*). We randomly chose females for offspring measures from the peak female emergence date for each subset of bee cells reared (*i.e.*, 8 weeks × six plots = 48 subsets) to standardize individuals among plots and weeks.

Frozen female bees were placed individually into vials, dried at 55 °C until weights stabilized, then reweighed to the nearest 0.1 mg on a Sartorius TE64 balance (Sartorius, Goettingen, Germany). To extract lipids, we added 10 ml of petroleum ether to each bee for 10 days, after which the ether was decanted and bees re-dried, weighed again, and proportion body lipid and total lipid mass calculated. To assess bee body size, we measured the head width of each bee (from the outer edges of the eyes) to the nearest 0.5 mm using a microscope fitted with an ocular micrometer (*O'Neill et al., 2011*; *O'Neill, Delphia & O'Neill, 2014*; *McCabe et al., 2021*). We chose this metric to be consistent with our previous work on ALCB body size and because it is a more easily obtained and less ambiguous measure than intertegular distance, another common metric for assessing bee body size (*Cane, 1987*). We (*O'Neill, Delphia & O'Neill, 2014*) and others (*McCabe et al., 2021*) have also shown that head width and intertegular distance are highly correlated with one another in ALCBs and are therefore comparable metrics for intraspecific comparisons.

***Estimating seed yields.*** In late fall, we randomly selected nine spatially stratified alfalfa plants in each plot, harvested all plants, and determined plant biomass and seed yields (measured as dry weight to the nearest 0.01 g).

## Statistical analyses: field study

We conducted all statistical analyses using JMP (Version 14; SAS Institute Inc., Cary, NC, USA, 1989–2023) (Data S1). All data were transformed as necessary to meet the assumptions of normality and homogeneity of variance. Alfalfa floral density, number of bee cells, number of completed nests, bee head widths, and plant weight were square-root transformed, proportion adult emergence success was arcsine transformed, and seed yield was ln transformed for the field study (2017). *Post-hoc* Tukey HSD tests were used to test for pairwise differences for significant effects, following each of the models described below.

***Alfalfa and wildflower floral density.*** To understand seasonal availability of food resources (alfalfa and wildflowers), we examined the effect of week on alfalfa floral density and wildflower density using separate mixed-effects linear models with week (ordinal) as a fixed effect and site as a random effect.

***Offspring production and survival.*** We investigated treatment effects on several measures of offspring production and survival including number of completed nests, number of bee cells, adult emergence success, and offspring mortality. Estimated number

of bee cells and number of completed nests provisioned per week per plot were highly positively correlated with each other (r = 0.98, $P < 0.0001$, $N = 48$); we report the results for number of bee cells only as a more easily understood metric of offspring production. We tested the effect of wildflower treatment on the number of bee cells produced and on adult emergence success using separate linear models with treatment and week (ordinal) as main effects and alfalfa floral density and bee shelter orientation as covariates. We tested the effect wildflower treatment on offspring mortality using a repeated-measures (life stage reached) MANCOVA with wildflower treatment and week (ordinal) as main effects and alfalfa floral density and bee shelter orientation as covariates.

*Offspring condition.* Following a significant MANCOVA testing the effect of wildflower treatment on head width and proportion body lipids of offspring (Wilks' 0.74, $F_{20,1500} = 11.95$, $P < 0.0001$), we performed separate linear models with treatment and week (ordinal) as main effects and alfalfa floral density and bee shelter orientation as covariates. Total lipid mass was positively correlated with proportion body lipids (r = 0.87, $P < 0.001$, $N = 762$) and we observed a similar response as proportion body lipids (Fig. S2, Text S1, Table S1); we report results for proportion lipids only.

*Estimating seed yields.* We tested the effect of flower strip treatment on alfalfa seed yields using a mixed-effects linear model with site as a random effect and plant biomass as a covariate.

**Cage study:** Do alternative wildflower resources enhance ALCB reproduction and offspring condition?

*Alfalfa plot.* In spring 2018, we initiated a field-cage study using one of the alfalfa plots from the field study (Fig. S3). We established 24 plots, each of which contained six alfalfa plants. Plots were randomly assigned to one of three food-resource (nectar and pollen) treatments: alfalfa only (A-only), alfalfa plus wildflowers (A+WF), or wildflowers only (WF-only) (see Fig. S3 for more details regarding plots and layout). All treatments contained alfalfa for nesting material; flower buds were removed from plants in the WF-only treatment so that no flowers were available for collecting pollen or nectar. Alfalfa and wildflower bloom overlapped.

For the A+WF and WF-only treatments, we selected four wildflower species from which bees collected pollen in the field experiment: *H. annuus*, *H. multiflora*, *P. campanularia*, and *P. tanacetifolia*. Each plot contained two *H. annuus*, two *H. multiflora*, and several rows of *P. campanularia* and *P. tanacetifolia* (Fig. S4). Wildflowers were planted in late May, and plots were watered every other day.

Each plot was enclosed in a separate cage comprised of a PVC-frame (1.5 m × 1.5 m × 2.5 m) covered with mesh netting (hole size 0.72 × 0.97 mm) (Green-Tek, Inc., Dinuba, CA) (*Slominski & Burkle, 2021*). In the northwest corner of each cage, we placed one wood-laminate block with 130 tunnels (5.5 mm in diameter) to provide nest sites for bees; tunnel openings faced southeast. Bee cells were obtained from Seed Source, Inc. on 1 June and reared (as above) for field-cage release. On 17 July we released 10 female and five male bees per cage and on 19 July we conducted night observations of the nesting blocks to ensure that female and male bees were present in each cage. After allowing 1 week for mating, we removed males from all cages to eliminate harassment by males of females.

Females foraged and nested for 6 weeks until 28 August, when we terminated the study. Due to the small size of the cages and the layout meant to maximize floral resources within the cages, we did not measure floral density to minimize disturbance of plants and bee foraging and nesting activities.

***Offspring production and survival among cages.*** In late August, all nest blocks were removed from cages. In late fall, all tunnels with cells were counted within each block and linear sequences of cells in each nest were broken apart into individual cells which were each placed into a labeled gelatin capsules (#0) with a pinhole in each end for air exchange. Cells were placed in cold storage at 6 °C on 5 November for overwintering. On 1 June 2019, we removed from cold storage all 2,293 gelatin capsules containing bee cells and reared offspring to adults in growth chambers at 28 °C. Emerging males and females were immediately frozen on the day of emergence. After adult emergences ceased, we counted the number of emerged adults, males, and females, determined adult sex ratios (males: females), and calculated emergence success of offspring as above. We dissected all cells from which no adults emerged to determine offspring stage of death, and, for each treatment, we calculated offspring mortality as above.

***Offspring condition.*** To evaluate offspring condition, we measured body lipid content and body size of adult female offspring as in the open-field study.

***Pollen provisioning and provision quality among cages.*** To assess the types of pollen provisioned by females and to relate pollen provision composition to provision quality and its effects on offspring condition, we removed frass from bee cocoons prior to adult emergence to identify the plant species that females visited to provision offspring. The A+WF and WF-only treatments allowed us to determine what bees were foraging on when they had a choice among species. In winter 2019, fecal samples (*i.e.*, frass) were collected from each bee cell with a cocoon, transferred to 1.5 mL microfuge tubes, and frozen. Later, we processed those fecal samples taken from cocoons from which female offspring emerged. To identify pollen, we added a solution of 95 μL deionized (DI) water and 5 μL 1% aqueous Safranin O solution (Home Science Tools, Billings, MT, USA) to each fecal sample. The samples were vortexed, allowed to sit for 4 h to soften the hardened frass, broken apart using a glass stirring rod, and then vortexed again. We then made pollen slides, identified the pollen types using light microscopy, and calculated pollen provision composition of fecal samples as above.

To assess provision quality, we determined the percent total nitrogen for the four wildflower species. Briefly, we collected pollen as anthers dehisced from each flower species from flower strips in the field study and sent samples to the Environmental Analytical Lab (MSU, Bozeman, MT, USA) for combustion analysis (Costech Elemental Combustion System 4010) to determine total nitrogen. We converted nitrogen values to protein content using a multiplier of 6.25 (*Buchmann, 1986*; *Roulston, Cane & Buchmann, 2000*). We used published values in the literature for the protein content of alfalfa pollen (*Stace, 1996*) due to difficulties harvesting alfalfa pollen. We used species-level values of percent pollen protein in conjunction with provision species composition to estimate the mean protein content (*i.e.*, quality) of each pollen provision. We excluded any pollen grains that we could not identify (≤7 per sample) from these calculations. We provide descriptive data on

the composition of alfalfa *vs.* wildflower pollen in provisions and the estimated pollen provision protein content (see Results); no statistical analyses were performed.

***Provision quality and offspring condition in A+WF cages.*** For the A+WF treatment only, we examined the relationship between the estimated protein content of provisions and body lipid content and body size of adult females.

## Statistical analyses: cage study

We conducted all statistical analyses using JMP (as above; Data S2). Sex ratios were ln transformed for the cage study to meet the assumptions of normality and homogeneity of variance. *Post-hoc* Tukey HSD tests were used to test for pairwise differences for significant effects, following each of the models described below.

***Offspring production and survival among cages.*** We investigated treatment effects on several measures of offspring production and survival including number of bee cells containing offspring, adult emergence success, numbers of adult male and female offspring, sex ratios, and offspring mortality. Following a significant MANOVA testing the effect of wildflower treatment on number bee cells, adult emergence success, number of adult male and female offspring, and sex ratios (Wilks' 0.13, $F_{10,34}$ = 5.89, $P$ < 0.0001), we performed separate one-way ANOVAs and analyzed each metric as 1) total per cage as an overall population measure that is of most relevance to seed growers and 2) the mean of cells per nest per cage as a per-capita (individual female) measure of biological relevance. We tested the effect wildflower treatment on offspring mortality using a repeated-measures (life stage reached) MANOVA and analyzed as 1) total proportion of offspring per cage and 2) the mean proportion of offspring per nest per cage. Because we observed similar responses for the two different measures for each of the six metrics, we report total per cage results only; see Supplement for mean per nest per cage results (Text S2; Tables S2 and S3).

***Offspring condition among cages.*** We tested the effect of wildflower treatment on head width and proportion body lipids of females using separate mixed-effects linear models with cage and nest (nested within cage) as random effects. Proportion body lipids and head width were not correlated with each other (r = 0.07, $P$ = 0.2229, $N$ = 327). Total lipid mass was correlated with proportion body lipids and head width (r > 0.66 for both, $P$ < 0.0001 for both, $N$ = 327) and we observed a similar response for total lipid mass (Fig. S5, Table S4); we report results for proportion body lipids and head width only.

***Provision quality and offspring condition in A+WF cages.*** Only in the A+WF treatment did bees have access to all the plant species used in all treatments, which could lead to variation in pollen protein content among provisions. Therefore, to investigate the potential effects of provision quality on offspring condition, we tested the relationship between pollen protein content and head width of females at emergence in the A+WF treatment only using a mixed-effects linear model with cage and nest (nested within cage) as random effects. Proportion body lipids (r = 0.17, $P$ = 0.0390, $N$ = 143) and lipid mass (r = 0.68, $P$ < 0.001, $N$ = 143) were correlated with head width and with each other (r = 0.80, $P$ < 0.001, $N$ = 143). We report results for head widths only; results for proportion body lipids and total lipid mass were similar (Table S5).

## RESULTS

**Field study:** Do late-season supplemental wildflower resources enhance ALCB reproduction and offspring condition?

*Alfalfa and wildflower floral density.* Alfalfa peak bloom preceded wildflower bloom as intended in our experimental design (Figs. 1A and 1B). Alfalfa floral density was highest in week 2 (week of July 10) and then declined in the following weeks to ca. 15% of the highest mean floral density during weeks 5–7 (week of July 31–week of August 14), and finally to ca. 3% in week 9 (week of August 28) ($F_{7,35}$ = 49.35, $P < 0.0001$). The wildflowers started blooming in week 5 (week of July 31) and then increased in the following weeks ($F_{7,14}$ = 85.19, $P < 0.0001$).

*Flower visitation and pollen load composition.* Female ALCBs at our sites were not often observed in the wildflower strips. During 8.95 h of observations, we recorded 16 female bees visiting wildflower strips on flowers of *P. campanularia* ($N = 7$), *P. tanacetifolia* ($N = 8$), and *C. officinalis* ($N = 1$). During 3.5 h of observations in vegetation surrounding our plots, we recorded eight ALCBs (seven females, one male) visiting hoary alyssum (*Berteroa incana* L.), six on flowers and two collecting leaf pieces.

We examined the pollen loads of 367 female bees returning to shelters in 2017. Pollen identifications revealed some anomalies, mainly that we observed what appeared to be *Phacelia* pollen in samples 3 weeks prior to *Phacelia* blooming in wildflower strips and we observed this for plots from both treatments. This indicated to us that (1) there may be other wild or cultivated plant species in this same genus within the broader landscape (*e.g.*, surrounding residential home gardens) that we were not aware of or (2) there may be another plant species within the landscape whose pollen is easily confused with *Phacelia* that was not in our reference collection and that we cannot account for. Therefore, we are not able to say from our pollen analyses whether bees collected *Phacelia* pollen from our wildflower strips. Additionally, only a single bee collected another pollen species which we identified as *Calendula*, possibly *C. officinalis*, from the wildflower strip. Consequently, we were limited to comparisons of alfalfa *vs.* non-alfalfa pollen.

Pollen loads from all plots and weeks combined ($N = 367$) consisted of a mean (± SE) of 81.1 ± 1.7% alfalfa pollen grains and 18.9 ± 1.7% non-alfalfa pollen grains. We identified non-alfalfa pollen grains in pollen loads at the onset of our bee-pollen collections (week 2), and throughout the nesting season. Two weeks after bee release pollen loads from all plots combined consisted of a mean of 29.7 ± 5.2% non-alfalfa pollen grains with a range for all plots from 19.9–48.3%. Alternative pollen resources used appeared to include those in the surrounding landscape, primarily a *Berteroa* type pollen, an *Achillea* type pollen, and three pollen morphospecies that were not in our physical reference collection.

*Offspring production and survival.* There was no effect of wildflower treatment on the total number of bee cells produced (Table 1). However, the number of bee cells completed varied by week peaking three weeks after release (Fig. 1C, Table 1). Most bee cell production (90%) occurred during the first five weeks after bee release. The remaining 10% occurred in weeks 6 (ca. 6%), 7 (ca. 2%), 8 (ca. 1%), and 9 (ca. 1%) after bee release when the wildflower strips were blooming. Bee cell production for any single plot ranged from

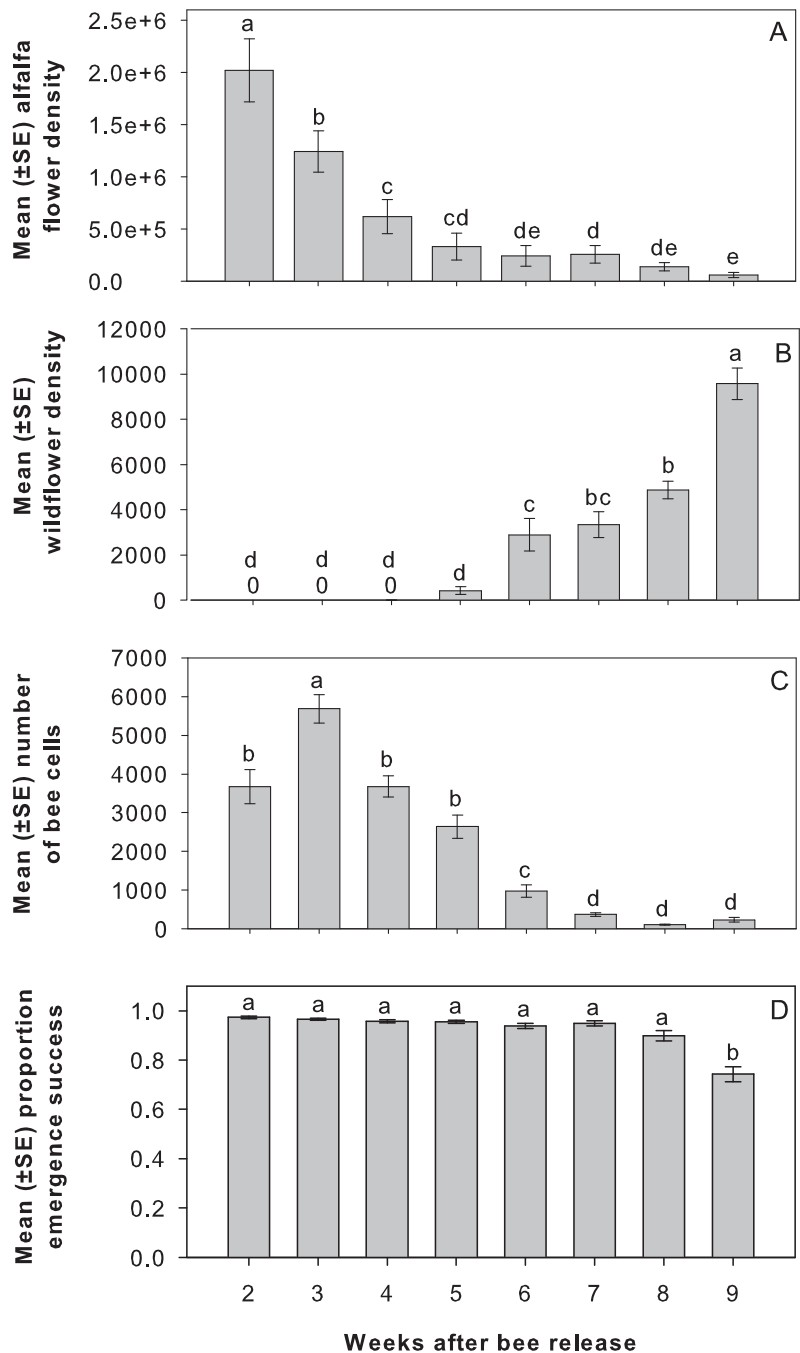

**Figure 1 Alfalfa flower density, wildflower density, number of bee cells, and adult emergence success by week in field study.** Mean (±SE) (A) alfalfa flower density, (B) wildflower density, and (C) number of bee cells produced each week after bee release in 2017; week 2 corresponds to the week of July 10, 2017. Mean (±SE) (D) proportion adult emergence success in 2018 from cells provisioned over 8 weeks in 2017. Note y-axes are at different scales. Lowercase letters above each bar indicate significant differences among weeks from Tukey's HSD test, $P < 0.05$.

**Table 1  Effects of wildflower treatment, week, alfalfa floral density, and shelter orientation on number of bee cells, adult emergence success, offspring mortality, and female offspring head width and proportion body lipids in field study.** Results of the influence of wildflower treatment, week, alfalfa floral density, and shelter orientation on number of bee cells produced in 2017, proportion adult emergence success in 2018 from cells provisioned in 2017, proportion of offspring that died at each of five life stages (*i.e.*, egg, larva, prepupa, pupa, and unemerged adult), and head width (mm) and proportion body lipids of adult female offspring that emerged in 2018 from cells provisioned in 2017. *P*-values in boldface are significant at α = 0.05.

| Source | df | Bee cells F | P-value | Emergence success F | P-value | df | Offspring mortality F | P-value | df | Head width (mm) F | P-value | Proportion body lipids F | P-value |
|---|---|---|---|---|---|---|---|---|---|---|---|---|---|
| Whole model | – | – | – | – | – | 10, 37 | 1.92 | 0.0739 | – | – | – | – | – |
| Treatment | 1, 37 | 1.03 | 0.3163 | 4.00 | 0.0530 | 1, 37 | 0.12 | 0.7324 | 1, 751 | 1.57 | 0.2099 | 0.10 | 0.7521 |
| Week | 7, 37 | 49.54 | **<0.0001** | 15.66 | **<0.0001** | 7, 37 | 2.27 | **0.0497** | 7, 751 | 7.38 | **<0.0001** | 6.23 | **<0.0001** |
| Sqrt (Alfalfa floral density) | 1, 37 | 0.11 | 0.7469 | 2.83 | 0.1009 | 1, 37 | 1.21 | 0.2794 | 1, 751 | 2.88 | 0.0901 | 12.39 | **0.0005** |
| Shelter orientation | 1, 37 | 5.53 | **0.0241** | 0.03 | 0.8557 | 1, 37 | 1.28 | 0.2644 | 1, 751 | 1.03 | 0.3104 | 3.58 | 0.0587 |

357–1,490 total cells provisioned during week 6, 220–536 cells during week 7, 68–148 cells during week 8, and 41–389 cells during week 9. Shelter orientation was negatively associated with bee cell production; shelters facing northwest were most associated with lower bee cell production, whereas those facing north (followed by northeast and east) were most associated with higher bee cell production.

Among just those cells in which an egg was laid, there was no effect of wildflower treatment on proportion of cells producing adult offspring after overwintering (Table 1). However, the proportion of cells producing adults varied by week of cell completion, declining in late summer (Fig. 1D, Table 1).

Among cells in which an egg was laid, but which did not produce live adults, there was no effect of wildflower treatment on offspring mortality (Table 1). Most premature deaths occurred at the larval stage with a mean (±SE) proportion of 0.34 ± 0.04 for treatments and weeks combined (N = 48). However, the week that cells were produced influenced the stage at which offspring died (Table 1). Neither shelter orientation nor alfalfa floral density was associated with offspring mortality.

***Offspring condition.*** Wildflower treatment did not affect head width or proportion of body lipids of adult female offspring that emerged in 2018 (Table 1). However, female offspring from cells completed earlier in the nesting season had greater mean head widths and higher proportion body lipids (Fig. 2, Table 1). Neither shelter orientation nor alfalfa floral density influenced head widths. Alfalfa floral resources were negatively associated with body lipids. Shelter orientation did not influence body lipids.

***Estimating seed yields.*** There was no main effect of treatment on alfalfa seed yields (*i.e.*, total seed mass) (Table S6). Greater plant biomass was associated with higher total seed mass.

**Cage study:** Do alternative wildflower resources enhance ALCB reproduction and offspring condition?

***Offspring production and survival among cages.*** Relative to the other treatments, females in the A+WF treatment provisioned more bee cells with an egg per cage

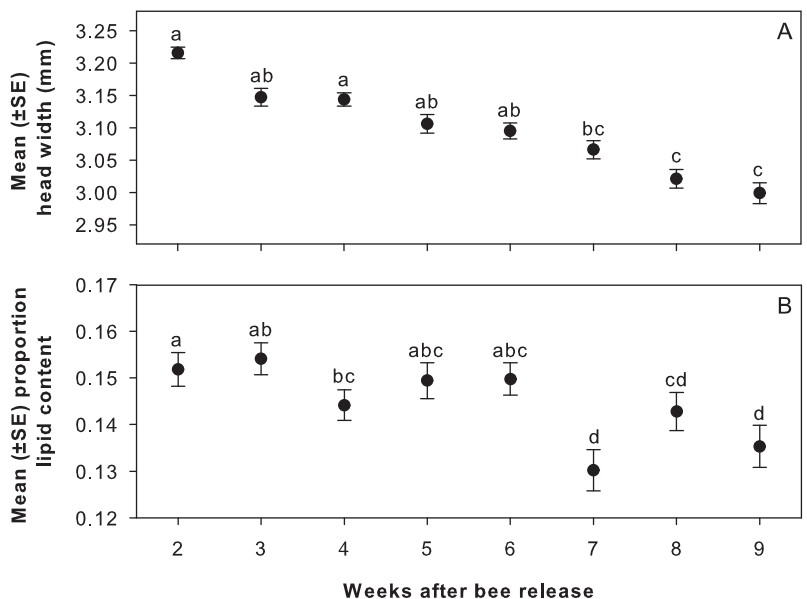

**Figure 2 Head width and proportion body lipids of adult female offspring by week in field study.** Mean (±SE) (A) head width (mm) and (B) proportion body lipids of adult female offspring that emerged in 2018 from cells provisioned over 8 weeks in 2017; week 2 corresponds to the week of July 10, 2017. Note y-axes are at different scales. Letters indicate significant differences among weeks from Tukey's HSD test, $P < 0.05$.

($F_{2,21}$ = 6.53, $P$ = 0.0062; Fig. 3A). Of those cells containing an egg, the proportion that produced adults was 17.9–26.8% higher in A-only compared to WF-only and A+WF ($F_{2,21}$ = 9.35, $P$ = 0.0012; Fig. 3B). More female ($F_{2,21}$ = 6.09, $P$ = 0.0082) and male ($F_{2,21}$ = 4.45, $P$ = 0.0245) offspring emerged per cage from A+WF compared to WF-only, and neither was different from the A-only (Fig. 3C). There was no difference among treatments in the mean male: female sex ratio per cage ($F_{2,21}$ = 1.06, $P$ = 0.3638).

Offspring mortality was similar across treatments ($F_{2,21}$ = 0.22, $P$ = 0.8067). Most premature deaths occurred at the larval stage (mean proportion all treatments combined = 0.33 ± 0.03, $N$ = 24); followed by prepupal, pupal, unemerged adult, and egg stages.

***Offspring condition among cages.*** Female offspring from A-only ($N$ = 115) emerged with significantly greater head widths ($F_{2,19.27}$ = 3.71, $P$ = 0.0433; Fig. 4A) and proportion body lipids ($F_{2,17.96}$ = 7.84, $P$ = 0.0036; Fig. 4B) than offspring from WF-only ($N$ = 67), but neither were different than offspring from A+WF ($N$ = 145).

***Pollen provisioning and provision quality among cages.*** Examination of pollen provisions in A-only cages confirmed that bees were foraging solely on alfalfa (mean = 98.8 ± 0.1%). In A+WF cages, provisions were comprised of alfalfa (mean = 73.9 ± 1.9%; range: 1–100 pollen grains), *Phacelia* spp. (24.4 ± 1.9%; range: 0–99), and *H. multiflora* pollen (0.5 ± 0.1%; range: 0–6). In WF-only cages, pollen provisions were comprised of *Phacelia* spp. (mean = 95.8 ± 0.7%; range: 70–100) and *H. multiflora* (2.9 ± 0.7%; range: 1–28). For all treatments, some pollen grains (≤7 in any one sample) were too distorted to determine their identity and were categorized as unidentifiable, which is why values do not sum to 100%.

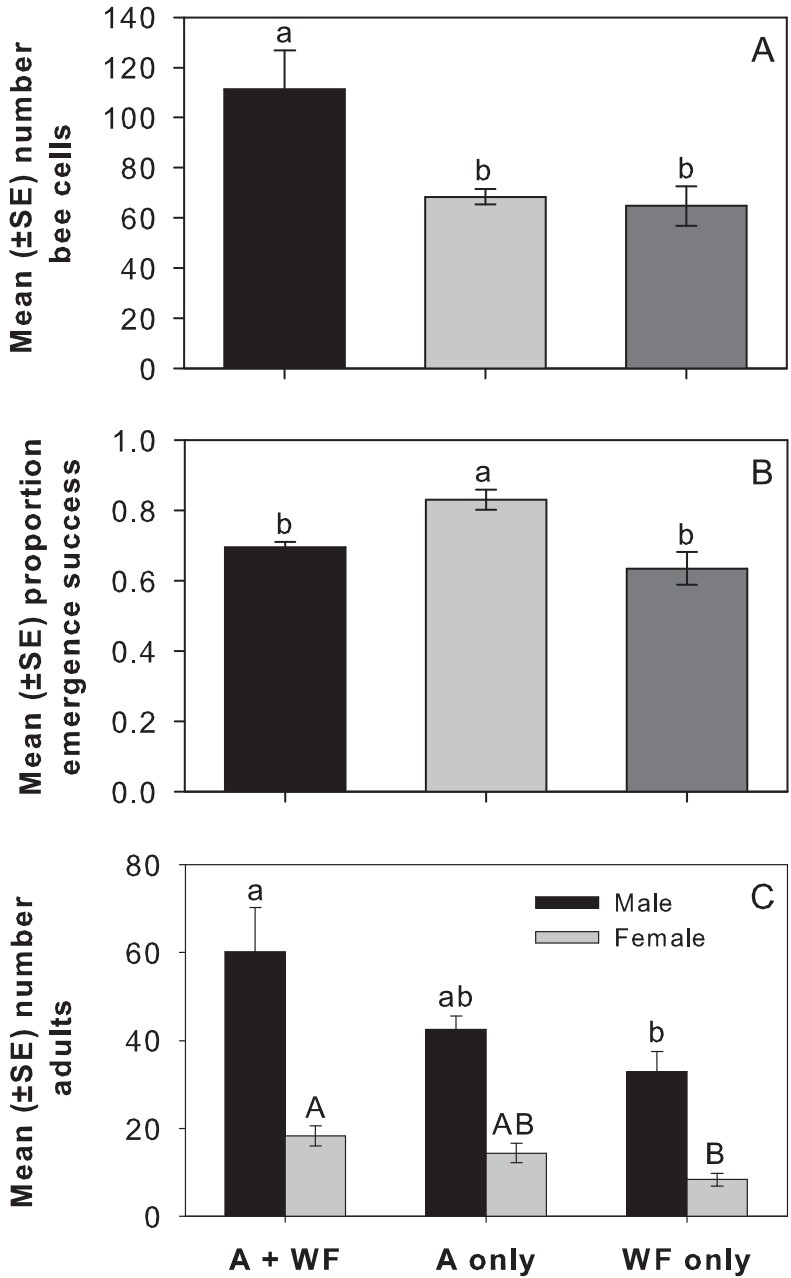

**Figure 3 Number of bees cells, adult emergence success, and number of adult bees by treatment in cage study.** Mean (±SE) (A) number of bee cells containing offspring (*i.e.*, an egg was laid), (B) proportion adult emergence success, and (C) number of adult bees per cage by treatment: alfalfa-plus-wildflowers (A+WF), alfalfa-only (A-only), or wildflowers-only (WF-only). Note y-axes are at different scales. Lowercase letters above each bar indicate significant differences between treatments from Tukey's HSD test, $P < 0.05$.

Estimated pollen provision protein content varied considerably among treatments, being highest in WF-only cells (mean = 55.5 ± 0.2%, N = 62) followed by A+WF (28.8 ± 0.7%, N = 143) and A-only (19.6%, based on *Stace (1996)*, N = 112).

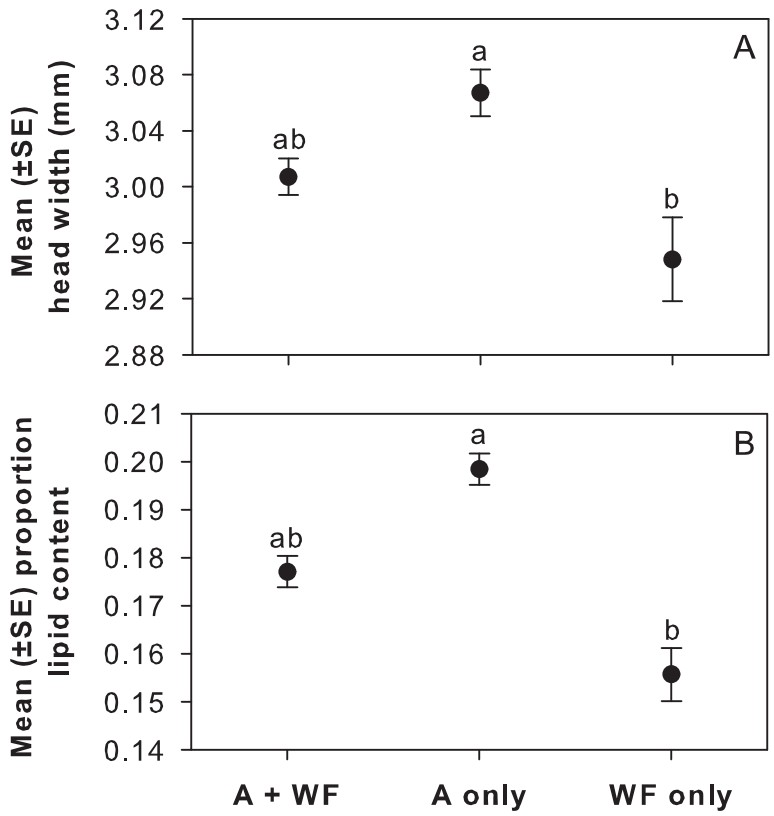

**Figure 4 Head width and proportion body lipids of adult female offspring by treatment in cage study.** Mean (±SE) (A) head width (mm) and (B) proportion body lipids of adult female offspring that emerged in 2019 from cells provisioned in 2018 by cage treatment: alfalfa-plus-wildflowers (A+WF), alfalfa-only (A-only), or wildflowers-only (WF-only). Note y-axes are at different scales. Lowercase letters indicate significant differences among treatments from Tukey's HSD test, $P < 0.05$.

***Provision quality and offspring condition in A+WF cages.*** Closer examination of the A+WF treatment revealed that pollen protein content was not significantly related to head widths of emerging adult females ($F_{1,141}$ = 2.04, $P$ = 0.1555).

# DISCUSSION

Research on the use of flower strips in agricultural landscapes for enhancing pollinators has focused primarily on two general questions: to what degree do flower strips (1) enhance pollination services in crops and (2) aid conservation purposes by supporting high wild bee diversity? Regarding bee conservation, there is ample evidence that wildflower plantings have positive effects on the abundance and richness of eusocial and solitary wild bees (*Blaauw & Isaacs, 2014*; *Williams et al., 2015*; *Campbell et al., 2017*; *Rundlöf, Lundin & Bommarco, 2018*; *Burkle, Delphia & O'Neill, 2020*; *Capera-Aragones et al., 2024*). However, few studies have directly assessed whether flower strips can increase bee reproduction (*Klatt, Nilsson & Smith, 2020*; *Ganser, Albrecht & Knop, 2021*) or the quality of offspring, both of which are important species-specific factors for increasing bee populations.

Our research focused on the reproductive success of a single species of solitary bee, the alfalfa leafcutting bee (ALCB) *M. rotundata*, that is often intensively managed for its pollination services, especially for alfalfa grown for seed production (*Pitts-Singer & Cane, 2011*). We combined open-field and field-cage studies to examine whether adding late-blooming floral resources in the form of wildflower strips would enhance the reproductive success of nesting females and the condition of their offspring. The results we present here add to our understanding of the relationships between floral resources and bee reproductive success, and are important for guiding other studies exploring floral resource provisioning strategies for supporting ALCBs and other solitary bees in agroecosystems.

### Do late-season supplemental wildflower resources enhance ALCB reproduction and offspring condition? The open field study

Our original goal was to replicate the open-field study during two summers, but delayed alfalfa growth in 2016 caused the alfalfa and wildflowers to bloom at the same time, leaving us unable to test our hypothesis that year. The opposite, though less crucial problem occurred in 2017 when alfalfa bloom declined rapidly and the wildflowers bloomed late, but we were able to compare the success of bees in the two treatments: alfalfa alone *vs.* alfalfa with adjacent wildflower strips. The data from the open-field study falsified our hypothesis that the addition of wildflower strips to seed alfalfa fields would increase the number of adult ALCB offspring and their condition. The results could be related to the fact that the peak of wildflower bloom occurred well after the highest density of alfalfa flowers had passed, leaving a mid-season gap of ca. 2 weeks in which neither alfalfa nor wildflowers in the strips were in full bloom. During this period, there were floral resources available, but they were limited because alfalfa was at ca. 15% of its highest bloom density. This decrease in food resources was likely due to some combined function of the phenology of the plant species we chose, the relative attractiveness of the wildflower species to ALCBs, and the drought conditions at the site that may have affected the quality of alfalfa and wildflowers late in the season; August 2017 was the second driest on record in our study area. There is also the question of whether the wildflower strips were large enough to attract females for repeated foraging trips. Ideally, in this system, wildflower strips would reduce the decline in total floral resources by blooming earlier as alfalfa flowers begin to senesce, allowing bees to continue provisioning brood cells with limited interruption in available food resources.

Although we observed few ALCBs visiting the wildflower strips (mainly the two species of *Phacelia*), our analysis of pollen loads collected from the abdominal scopae of females showed that they regularly visited plant species other than alfalfa; we could not be sure how much of the non-alfalfa pollen collected came from the wildflower strips due to unidentified plant species in the landscape with pollen similar to our *Phacelia* species. Indeed, bees collected non-alfalfa pollens during the second week after release, at a time when wildflower strips were not in bloom. Thus, even before the alfalfa bloom declined, females clearly foraged in the landscape beyond our plots where other flowering plants were quite sparse. For all plots and weeks combined, 19% of pollen grains carried by

females were from non-alfalfa plants, including what appeared to be primarily an *Achillea* type pollen (Asteraceae) and a *Berteroa* type pollen (Brassicaceae), likely *B. incana* which we recorded bees visiting during timed observations, as well as three species we could not match to our reference collection. Such a diversity of pollen sources is not unusual, even for ALCBs nesting amid commercial alfalfa fields. At another site in southcentral Montana, several studies showed that the plant species composition changed and volume of scopal pollen loads declined as the nesting season progressed, with the bees adding pollen from other plant species, especially mustards (Brassicaceae) (*O'Neill et al., 2004*; *O'Neill & O'Neill, 2011*). Our study provides further evidence that ALCBs use other plant resources even when nesting in alfalfa fields. The value of alternative pollens as food resources requires further examination.

One limitation of our field study is the lack of temporal replication. Despite designing and conducting 2 years of field experiments, we ended up with a single year due to previously mentioned difficulties. In addition, this study was conducted at a single location though simultaneous replication at other localities with different climates and seasonality of bees would be logistically difficult. Lastly, our constraints on floral selection may have been too stringent and the use of favorable forage plants for *M. rotundata* such as those in the family Fabaceae (*e.g.*, *Horne, 1995*) as a positive control, and/or a larger, more diverse suite of plants from additional plant families including Brassicaceae could have benefited the overall study design. Further research should include additional plant species not tested directly in this study. Additional study examining the usefulness of this flower mix as food for wild and managed bee species is also warranted (see *Burkle, Delphia & O'Neill, 2020*) as this diverse, late-blooming wildflower mix provided resources for more than 25 wild bee species in the landscape (CMD personal obs. 2017), including a bumble bee species of concern *Bombus occidentalis* Greene (*Janousek et al., 2023*).

### Do alternative wildflower resources enhance ALCB reproduction and offspring condition? The cage study

Because we could not be certain how much of the non-alfalfa pollen came from the flower strips in the open-field study pollen analyses, we initiated a cage study to further examine alternative floral resource effects on ALCB reproductive success and offspring condition in a controlled setting. In the A+WF and WF-only cages, females collected wildflower pollen and produced offspring that reached the adult stage the following year. However, adding wildflowers as alternative food resources in A+WF cages did not provide significant benefits for bees beyond the A-only treatment. Similar numbers of adult male and female offspring were produced on mixed and A-only diets, and females were of similar size and proportion body lipids. Furthermore, though bees in A+WF provisioned more cells, the survival of offspring to adults was highest in A-only cages. WF-only treatments also produced similar numbers of adult male and female offspring as A-only diets, but females were smaller with lower proportion body lipids, despite the higher protein content of wildflower than alfalfa pollen. Thus, protein content alone does not affect offspring condition, particularly when alfalfa resources are also available.

Pollen is the most important food resource for larval development and female egg production (*Dobson & Peng, 1997*) and pollen from different plant species varies in nutritional content (*Buchmann, 1986*; *Roulston, Cane & Buchmann, 2000*; *Willmer, 2011*; *Danforth et al., 2019*; *Vaudo et al., 2020*). The results of the cage study may indicate that pollen protein content, as well as other nutritional components of alfalfa pollen, were already above some necessary minimum in all treatments as long as a sufficient mass of provisions was provided to offspring. However, there may be some other essential nutrients missing in our wildflower mix that alfalfa provides. Other nutritional factors associated with pollen that affect bee development include amounts of particular amino acids, lipids, starches, and sterols, as well as protein to lipid ratios (*Roulston & Cane, 2000*; *Williams, 2003*; *Vaudo et al., 2020*).

One limitation of our cage study is that, due to logistical constraints of cage size and plant layout, we did not measure floral density in cages. We also did not measure the quantity of pollen in each provision due to the need to allow offspring to feed and complete development. Therefore, we cannot separate the roles of pollen quantity and quality, though they likely interact to affect overall nutrition and ALCB reproductive success. It is also possible that differences in nectar quality or quantity may have affected reproduction and offspring condition (*Burkle & Irwin, 2009*). Treatments might have also differed in ways other than available food resources. Because we consistently cut alfalfa plants back to keep them from blooming in the WF-only cages, we could have unintentionally affected the abundance or quality of alfalfa leaves available for nesting between treatments. Microclimate could have also differed based on the varying amounts and growth habit of vegetation and bare ground in cages.

## Seasonal effects on offspring condition

Although we found no benefit of wildflowers, we did observe a clear seasonal effect on reproductive success in the open-field study that has implications for ALCB management. Upon eclosing in 2018, female offspring from nests provisioned earlier in the season were significantly larger and had higher proportion body lipids compared to those provisioned later when alfalfa flower density was low, and the wildflower strips had not yet reached full bloom. This seasonal effect persisted in offspring from this study even when they were reared after an additional 1.5–2.0 years of winter storage under different thermal regimes; late-season offspring had lower winter survival and adult body mass (*Park et al., 2022*). The decline in size and lipid content of late-season offspring could be due to reduced alfalfa floral resources near nests, forcing females to forage further away and decreasing pollen-load size (*Peterson & Roitberg, 2006a*). This may account for female ALCBs returning to nests with smaller pollen loads later in the season in another study (*O'Neill & O'Neill, 2011*).

Heritability for body size is close to zero in ALCBs (*Owen & McCorquodale, 1994*), so variation among offspring is heavily influenced by the amount of provision provided by nesting females (*Klostermeyer, Mech & Rasmussen, 1973*; *Kim, 1999*; *Peterson & Roitberg, 2006b*; *Bosch, 2008*; *Danforth et al., 2019*; also see *Roulston & Cane, 2002*). Therefore, we cannot discount the idea that the much of the decrease in size of offspring produced later

in the season is a maternal effect, one that could reduce the success of daughters eclosing the following year. As nesting female ALCBs age, they also experience increased wing wear (*O'Neill, Delphia & Pitts-Singer, 2015*), which is known to increase mortality in bumble bees and affect their flower choices during foraging (*Cartar, 1992*; *Foster & Cartar, 2011*). Lipid stores of female ALCBs also quickly decline with age following eclosion (*O'Neill, Delphia & Pitts-Singer, 2015*). Reductions in the size of offspring in late-season cells has also been attributed to declining maternal condition in studies of several species of Megachilidae of the genus *Osmia* (*Tepedino & Torchio, 1982*; *Seidelmann, 2006*). Thus, if maternal aging rather than declining floral resources is the main cause of reductions in offspring quality, then adding wildflower strips may not greatly aid in increasing reproductive output in managed populations of ALCBs. Our data cannot separate the two hypotheses.

Whatever the cause, small daughters are likely to pay fitness costs that could in turn be passed on to their own daughters, perhaps leading to multiple generations of females of low quality in particular genealogical lineages. As in many insects, female body size in solitary bees is positively correlated with traits that can affect reproductive success (*Honěk, 1993*; *Bosch & Vicens, 2006*; *Rehan & Richards, 2010*; *Seidelmann, Ulbrich & Mielenz, 2010*). Studies have shown that larger female solitary bees can provision a cell in a shorter time, provision more cells, provide greater investment in individual offspring, produce more female offspring, produce larger eggs, and emerge with greater lipid stores (*Larsson, 1990*; *Sugiura & Maeta, 1989*; *Tengö & Baur, 1993*; *Kim, 1997*; *Rehan & Richards, 2010*; *Seidelmann, Ulbrich & Mielenz, 2010*; *O'Neill, Delphia & O'Neill, 2014*). Stored lipids are important for egg production in insects, and lipids (*Arrese & Soulages, 2010*) have been shown to quickly decline 1 week after adult emergence in ALCBs, a time that coincides with the period of terminal oocyte maturation just before the first eggs are laid (*Richards, 1994*; *O'Neill, Delphia & O'Neill, 2014*; *O'Neill, Delphia & Pitts-Singer, 2015*). It has been hypothesized that larger eggs with greater nutrient stores lead to reduced larval mortality and larger adult offspring (*Larsson, 1990*).

Lower levels of investment in offspring can lead to a second problem for sustaining populations of ALCBs: sex ratios skewed toward males. In *Megachile apicalis* Spinola, when resource levels were halved in a cage study, females provisioned fewer cells per day, invested less in female offspring, and skewed sex ratios towards males (*Kim, 1999*). In some instances, under very low resource levels, some solitary bee species may produce no female offspring at all (*Slominski & Burkle, 2021*). We did not examine sex ratios in the field study, however, in the cage study, there was no significant effects of treatment on sex ratios of successfully eclosing offspring, suggesting that resources among cages were favorable enough (by some measure) for nesting females to invest in producing female offspring; this warrants further investigation of the value of the two *Phacelia* species as food resources when pollen quantities are controlled.

## CONCLUSIONS

Our study was motivated by the general interest in using wildflower plantings to support bees in agricultural systems and by the specific problem that some alfalfa seed growers face
sustaining ALCB populations without purchasing replacements each growing season. With our experimental design in the open-field study and with our choice of wildflower species, however, the supplemental late-season floral resources did not enhance ALCB reproduction. In addition, female offspring produced later in the season emerged the following year with smaller body sizes and with lower lipid stores compared to those produced earlier in the season when alfalfa was in full bloom. Those two effects may place constraints on the reproductive success of those offspring and, perhaps, their value as pollinators. If this seasonality effect was due, in part, to maternal aging rather than just the decline in alfalfa resources, it may be that late-blooming wildflower strips will have limited value in sustaining healthy ALCB populations. Parallel results in our cage study revealed that the availability of both wildflowers and alfalfa did not result in any measurable reproductive benefits beyond alfalfa only. The availability of wildflowers in addition to alfalfa did not produce more offspring, larger offspring, or increase their lipid stores. Furthermore, when only wildflowers were available in the cage study, it resulted in smaller female offspring with lower lipid stores compared to those on alfalfa. If this effect was the result of pollen quality and not simply the amount of pollen available in WF cages, it could exacerbate the reproductive success of the already reduced condition that we observed for adult female offspring produced late in the season the previous year.

Our results highlight the importance of measuring multiple metrics of bee reproductive success, since using only one metric could lead to erroneous conclusions. For example, one might conclude that WF-only diets are not different from A-only diets if one examined only offspring numbers. But, if offspring size and body lipid content are important for future reproductive success and pollinator efficacy, the WF-only diet we chose may provide limited benefits to the long-term health and fitness of the population. To better guide future floral resource provisioning strategies, more research is needed to understand how alternative floral resources and nutrition, particularly pollen quality and quantity, and the temporal availability of those resources affects the reproductive success of ALCBs (and wild bees more generally), as well as the success of subsequent generations.

## ACKNOWLEDGEMENTS

We thank Tom Helm (local MT alfalfa seed grower; Seed Source, Inc.), Dave Gettel (MSU Farm Operations Manager), and Robert Dunn and Laura Smith (formerly Westscape Nursery) for field logistical support and helpful advice on establishing alfalfa plots and wildflower strips; Justin Runyon, Ruth O'Neill, Amelia Dolan, Lisa Seelye, Lauren Larios, and members of the Burkle Lab (MSU) and the Peterson Lab (MSU) for help planting alfalfa plots; Ruth O'Neill for designing and building bee shelters; Anthony Slominski, Perry Miller, and Jeff Holmes for use of equipment; and field and laboratory assistants Jacklynn Lathrop, Frances Ambrose, Alissa Freeman, Simone Durney, Amanda Yetter, and Bonnie Wolgamot. We also thank reviewers Kelsey Graham and Michael Killewald for providing helpful comments that improved the manuscript.

## Funding

This work was supported by the Agriculture and Food Research Initiative (No. 2016-67013-24752) from the USDA National Institute of Food and Agriculture, the Montana State University College of Agriculture Research Innovation Grant Program (No. 911041), and the Montana Agricultural Experiment Station. The funders had no role in study design, data collection and analysis, decision to publish, or preparation of the manuscript.

## Grant Disclosures

The following grant information was disclosed by the authors:
Agriculture and Food Research Initiative: 2016-67013-24752.
Montana State University College of Agriculture Research Innovation: 911041.
Montana Agricultural Experiment Station.

## Competing Interests

The authors declare that they have no competing interests.

## Author Contributions

- Casey M. Delphia conceived and designed the experiments, performed the experiments, analyzed the data, prepared figures and/or tables, authored or reviewed drafts of the article, and approved the final draft.
- Laura A. Burkle conceived and designed the experiments, performed the experiments, analyzed the data, authored or reviewed drafts of the article, and approved the final draft.
- Joshua M. Botti-Anderson performed the experiments, authored or reviewed drafts of the article, and approved the final draft.
- Kevin M. O'Neill conceived and designed the experiments, performed the experiments, prepared figures and/or tables, authored or reviewed drafts of the article, and approved the final draft.

## Data Availability

Raw data are available in the Supplemental Files.

## Supplemental Information

Supplemental information for this article can be found online at http://dx.doi.org/10.7717/peerj.17902#supplemental-information.

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
