# Peer review of "Seasonality and alternative floral resources affect reproductive success of the alfalfa leafcutting bee, Megachile rotundata"

_PeerJ, doi:10.7717/peerj.17902_

## Round 0.1 · original submission · Minor Revisions

Thank you for your submission. The reviewers have both recommended minor revisions which I believe will strengthen the final manuscript. I look forward to receiving the editted manuscript with these recommendations considered.

·

Basic reporting

L92: "similarly late bloom" seems repetitive. Consider removing.
L108: What about benefits for 2nd generation M. rotundata? Perhaps you don't get a second generation in Montana?, but it's relatively common throughout much of alfalfa seed production in the west, and seems like this could be a way to support the 2nd generation instead of it being a reproductive dead end, which is often the case when there isn't sufficient alfalfa in bloom still when they emerge late in the summer.

Experimental design

L204: Can you report how many bees were released per plot? Were bees released at the shelters?
L206-214: I think "haphazardly" would be more accurate than "randomly"
L221: How did you measure floral density?
L395 and L428: Should this be A+WF, instead of A+FS?

Validity of the findings

Results throughout - this is somewhat personal preference, but I would prefer to see the stats reporting in text where most relevant, and then the reporting tables included in the supplemental. So many tables feels a bit cumbersome.

L476: Which shelter orientation was most associated with negative bee cell production?

Additional comments

I enjoyed reading this manuscript! It was a well designed study (sorry the first year didn't work out! Such are field studies), and a well written manuscript.

·

Basic reporting

The authors designed and implemented a field study to test the effects of a selected wildflower mix on M. rotundata offspring provisioning, pollen collection, and developmental success. Overall, the manuscript is well prepared and I have few grammatical corrections (see edits by line). I do find some limitations of their study that are not properly addressed (see edits by line and also below) but overall, I see no reason why this manuscript could not be published with minor corrections- provided the comments are addressed.

Experimental design

Although the authors only collected data at one site for one field season for each experiment, they collected an impressive dataset and tested several metrics to understand the responses of M. rotundata to the wildflower mix. Overall, the study was conducted in such a way to test their hypothesis given the constraints that the authors described (utilizing plants that bloom after alfalfa and do not produce seeds that could reduce alfalfa seed quality through contamination). Though these constraints are valid for alfalfa seed production, I find that they are a limitation of the study that needs to be discussed further. Given the controlled nature of the experimental plots, it is unfortunate that the authors did not include at least one treatment with flowers commonly utilized as alternative forage for M. rotundata (such as Melilotus, Trifolium, and/ or Brassicaceae). I find no major flaws with their experimental design outside of the floral selection, though the lack of spatial and temporal replication should be highlighted in the discussion as an additional study limitation.

Validity of the findings

Their findings appear to be valid given the constraints highlighted by the authors. Though, I find the identifications of some pollen types are almost certainly too confident and should be more generalized (see comments by line for more details). I doubt the identifications of pollen to the species level, especially given that the authors had several pollen types that were not identifiable simply because they were not included in their reference collection. The authors should have referenced outside guides in addition to their own reference collection as several exist (Sawyer- “pollen identification for beekeepers”, global pollen project, PalDat, Isaacs lab Flickr page, etc.) and would provide more validity to their identifications. Additionally, information about those who helped to identify the pollen samples is important to understand the skillset and associated validity of the identifications (this is not a huge deal here since the authors primarily report alfalfa/non-alfalfa pollen, but is a good practice nonetheless).

Additional comments

Overall comments
Regarding the field study experiment:
1) Previous work, cited by the authors, has found that M. rotundata uses mainly Brassicaceae as an alternative forage, yet their wildflower mix does not include any members of this family (L157). The wildflower mix uses mostly Asteraceae. Given the authors objectives to select plant species that flower after alfalfa bloom, I understand their use of Asteraceae. However, this limitation needs to be discussed as most research seems to show that M. rotundata does not prefer to use Asteraceae pollens. Given the gap between alfalfa and the wildflower bloom coupled with selecting plants that are generally not utilized as much for foraging, it is rather unsurprising that M. rotundata did not benefit much from the wildflower mix. Thus, these limitations need to be discussed further.
2) Most studies find that M. rotundata uses Melilotus pollen as an alternative forage. I understand the need to minimize weed seeds within the adjacent fields, but in a controlled study, the authors should have included at least one treatment with favorable forage plants for M. rotundata. Have any past studies found that the addition of Melilotus (or other forage plants that were excluded as they have similar seeds to alfalfa) benefit M. rotundata reproductive success or output?
3) The field study experiment was conducted at one site in one sampling year. Although the authors collected a large amount of data, the lack of spatial or temporal replication here needs to be discussed as a limitation of the study.
a. Do the authors have pollen data from the previous year of the study where alfalfa and wildflower bloom overlaps? This could be used to support (or refute) the argument that wildflowers that bloom after crop bloom need to be utilized in alfalfa seed production plots depending on the use of non-alfalfa pollens for foraging during this time of resource overlap.
4) The use of head width and not inter-tegular distance to measure bee body size should be supported with citations as this does not seem to be common practice.
5) Nonetheless, their results are still important and valid, but the above points should be discussed further as they are large limitations of this study.

Regarding the field cage study
Maybe I missed it somewhere, but did the wildflower and alfalfa bloom overlap in the field cage experiments or was there a large gap in resources as seen in the field experiment?




Edits by line are below

Abstract
“prior to wildflower bloom .” extra space

Introduction
L54: suggest adding a sentence or two describing issues with diseases as well
L90-91: highlight most important alternative pollen species or groups found in the O’Neil and O’Neil (2011) study
L89-90: the study found a high proportion of Brassicaceae in M. rotundata pollen loads. Was canola planted at any nearby fields? If so, then restructure sentence to remove “weeds” since canola would be a likely source of this pollen type
L189-190: how far were nesting shelters from the strips?
L215-216: generally speaking, were alfalfa flowers evenly distributed throughout the plots or patchy?
L221: when did the floral strip begin to bloom and when did alfalfa peak/ finish blooming during your sampling year?
L228-230: were individuals netted or visually noted and recorded? If the latter, were measures taken to not record the same individual more than once?
L230-233: how many species of plants were samples in the surrounding vegetation? Presumably few, but this information helps readers to understand the availability of resources in the surrounding area.
L238-240: how many species were included in the reference collection? From how many botanical families? A small reference collection makes identifications difficult. Were any additional guides referenced for pollen types that were not included in the collection? (Sawyer- pollen identification for beekeepers, PalDat, global pollen project, etc.) Additionally, how many different people were involved with identification of pollen grains? There can sometimes be discrepancies in identifications between individuals, so it would be best if one (or as few people as possible) identified the samples. The authors described pollen mainly as “alfalfa or non-alfalfa”, which lowers the likelihood of misidentifications, but this information is still important to include nonetheless.
L240-142: were percentages calculated based on the number of grains or total volume of grains? In other words, would 5/100 smaller grains and 5/100 larger grains each constitute 5% of the total sample, or would percentages be corrected for size of pollen grains?
L280: why was bee head width used as a measure of body size and not inter-tegular distance as commonly suggested by others?
L288: were linear models or ANOVAS used prior to the Tukey HSD test?
L386: were the pollen frass samples identified using light microscopy?
L395: elsewhere you refer to this treatment as A+WF, is A+FS a separate treatment? Double check for consistency
L399: again, were ANOVAS or linear models used prior to the post hoc test?
L420: similar note to L280 and only needs to be described once, but why was head width and not IT distance selected as a metric for body size?
L428: check A+FS


Results
L471 and elsewhere: Though discussion of potential pollen species is welcomed, the confident identifications of pollen to species (especially Asteraceae) using light microscopy (assuming they used this method as it is not stated explicitly) is often impossible. For Asteraceae, even identifications to genus are often unreliable. Thus, using “type groups” as described by Sawyer in “pollen identification for beekeepers” or another similar guide is better for common problematic groups (such as Asteraceae and some Fabaceae) as one cannot be certain that the grains are even from Achillea. I believe that Graham et al. (2020) found several pollen types in Megachile pollen loads that were not located during their floral surveys, so it is likely that Megachile in this study foraged outside of the plots and could have found other species than yarrow.

The authors should not list species level identifications for pollen just because they match reference slide in their collection as most members within a genus will have similar pollen structure. Identifications should be listed to the lowest confident taxonomic level possible, which is often genus level but depends on the skill set of the person identifying the grains.

L476-477: how so? Which directions contained more cells?
L474: how many cells were produced during wildflower bloom? It is possible that cell production had almost stopped by the time the wildflowers even started to bloom
L489-494: interesting that cells completed earlier in the season had a higher lipid content, where alfalfa flowers were dominant, while alfalfa floral density had a negative effect on lipid content.
L518: cages should not have any alternative forage, so although 1.2% is very low (and could even be due to contamination) what were the other forage species? Were they unidentifiable? If so, state that here to be clear.

Discussion
L364-365: how long was the gap in resource availability?
L582-583: see note above on taxonomic identifications of pollen grains. Speculation of species in the discussion is fine, but reporting that they used specific species (yarrow and hoary alyssum) when identifications are performed under light microscopy (assuming that is the case) is erroneous- especially if the authors have unidentifiable species of pollens from their samples.
L682: further research should also include additional plant species not tested directly in this study
L689-692: have any past studies utilized differing planting dates of alfalfa to create a longer season of “full bloom” to test the resources abundance v. maternal aging idea discussed here? If so, would be good to include somewhere.
L716: alfalfa seed grower?

---

## Round 0.2 · accepted · Accept

Many thanks for your thorough treatment of the two reviewers comments. I am happy to suggest the journal accept that manuscript. Congratulations